# Zipf’s Law of Vasovagal Heart Rate Variability Sequences

**DOI:** 10.3390/e22040413

**Published:** 2020-04-06

**Authors:** Jacques-Olivier Fortrat

**Affiliations:** UMR CNRS 6015 Inserm 1083, Centre Hospitalier Universitaire Angers, 4 Rue Larrey CEDEX 9, 49933 Angers, France; jofortrat@chu-angers.fr

**Keywords:** baroreflex, heart rate variability, self-organized criticality, vasovagal syncope, Zipf’s law

## Abstract

Cardiovascular self-organized criticality (SOC) has recently been demonstrated by studying vasovagal sequences. These sequences combine bradycardia and a decrease in blood pressure. Observing enough of these sparse events is a barrier that prevents a better understanding of cardiovascular SOC. Our primary aim was to verify whether SOC could be studied by solely observing bradycardias and by showing their distribution according to Zipf’s law. We studied patients with vasovagal syncope. Twenty-four of them had a positive outcome to the head-up tilt table test, while matched patients had a negative outcome. Bradycardias were distributed according to Zipf’s law in all of the patients. The slope of the distribution of vasovagal sequences and bradycardia are slightly but significantly correlated, but only in cases of bradycardias shorter than five beats, highlighting the link between the two methods (r = 0.32; p < 0.05). These two slopes did not differ in patients with positive and negative outcomes, whereas the distribution slopes of bradycardias longer than five beats were different between these two groups (−0.187 ± 0.004 and −0.213 ± 0.006, respectively; p < 0.01). Bradycardias are distributed according to Zipf’s law, providing clear insight into cardiovascular SOC. Bradycardia distribution could provide an interesting diagnosis tool for some cardiovascular diseases.

## 1. Introduction

Complexity, the final frontier of the cardiovascular system, has emerged as a major topic over the last decade. Complexity was initially discovered from incidental findings when studying cardiovascular variability. The meaning and implications of these findings have long remained unclear. Cardiovascular complexity has since been more precisely described over time. It became more and more difficult to integrate it into the current view of cardiovascular physiology that is largely dominated by the deterministic homeostatic principle. According to this principle, physiological variables are regulated and maintained at their normal values thanks to negative feedback regulatory loops. Today, complexity challenges the homeostatic view on the cardiovascular system [1,2,3]. We recently demonstrated that at least part of this complexity is explained by the self-organization of a cardiovascular system poised at criticality [4,5]. We showed that occurrences of spontaneous vasovagal events are distributed according to Gutenberg Richter’s law. This law has been initially described in earthquakes occurrences: the magnitude plotted against the total number of earthquakes of at least this magnitude draws a straight line on a log-log graph. This finding explained how vasovagal reaction may occur. Vasovagal reaction is a parallel bradycardia and decrease in blood pressure of varying intensity from self-limiting symptoms to loss of consciousness and prolonged postictal asthenia [6]. During vasovagal syncope, the blood pressure decrease is not compensated by an increase of the heart rate as expected due to blood pressure homeostatic regulatory mechanisms. Brain perfusion is compromised because of the blood pressure decrease, and loss of consciousness eventually occurs. The self-organized pathophysiology of vasovagal syncope is a major finding, but its implication for cardiovascular physiology in general remains limited. Self-organized criticality has, however, emerged as a major topic in the study of dynamical systems and as a unifying theory across science fields, including physics, chemistry, ecology, and biology [1,2,7,8]. A better understanding of its meaning and implications for the cardiovascular system is needed. The study of cardiovascular self-organized criticality through vasovagal events requires continuous beat-by-beat recordings of blood pressure and heart rate. These recordings are difficult to obtain in some environmental conditions and are limited by time. These difficulties are a barrier toward a better understanding of cardiovascular self-organized criticality. Zipf’s law has initially been described based on word occurrence in a text: the frequency of any word in a text is inversely proportional to its rank of occurrence [7,9]. This law has been inscribed into beat-by-beat recordings of the heart rate (Heart Rate Variability, HRV). These recordings show a linear distribution of occurrence of non-specific consecutive heart rate sequences across several beats, these sequences being the “words” of the cardiovascular system “language” [10,11,12]. Zipf’s law represents another argument for cardiovascular self-organized criticality but without physiological and medical implications, contrary to Gutenberg Richter’s law [2,4,10]. Beat-by-beat recordings of the heart rate are easy to obtain by means of commercially available heart rate monitors, facilitating the study of its complex dynamics [13,14]. However, it is still unknown whether Gutenberg Richter’s law, determined by blood pressure and heart rate recordings, and Zipf’s law, determined only by heart rate recordings, provide the same information. The goals of our study were to check, first, whether Zipf’s law is observed specifically in bradycardia sequences, and second whether the meaning of Zipf’s and Gutenberg Richter’s laws overlap.

## 2. Materials and Methods 

### 2.1. Patients

This study focused on patients with a history of iterative vasovagal syncope. Patients and flow charts have previously been extensively described [4]. One hundred consecutive patients who came to our department for advice on their iterative loss of consciousness and who gave their informed consent were included (51 female, 43 ± 2 years, 1.67 ± 0.01 m, 68 ± 1 kg, mean ±Standard Error of the Mean, SEM). Thirty patients were excluded because their interview was not suggestive of vasovagal syncope or because of a history of heart disease. A detailed medical history is central to the diagnosis of vasovagal syncope, but the head-up tilt test may help in both diagnosis and management. The head-up tilt test identified three patients with an orthostatic hypotension and five patients with a postural tachycardia syndrome. These eight patients were excluded. From the remaining 62 patients, 34 had a positive outcome to the head-up tilt test with (near) syncope symptoms, and 24 of them could be paired in age and sex with patients with a negative outcome. The group of patients with a positive outcome was called T+ patients (16 female, 39 ± 3 years, 1.66 ± 0.01 m, 67 ± 2 kg). The group of patients with a negative outcome vas called T- patients (16 female, 39 ± 3 years, 1.69 ± 0.02 m, 69 ± 3 kg). Patients received a complete description of the experimental procedure before giving their written informed consent. The Comité Consultatif de Protection des Personnes dans la Recherche Biomédicales des Pays de la Loire (Regional Committee for the Protection of Persons, #00/08, May 30th, 2000), France, approved the experiment, which is in accordance with the declaration of Helsinki, Finland. 

### 2.2. Head-Up Tilt Table Test

The head-up tilt table test was performed in a quiet room with a comfortable ambient temperature (22–24 °C). The patient was lying on the table for at least ten minutes of adaptation to the supine position. The test began after 10 min in the supine position, and was followed by a 45 min period in the head-up position at an inclination of 70° by means of a motorized inclination table (AkronA8622, Electro-Medical Equipment, Marietta, GA, USA). The head-up position was stopped before 45 min elapsed in the event of (pre) syncopal symptoms defining the positive outcome. Cardiovascular monitoring was performed during the whole test by means of an electrocardiogram and a digital blood pressure monitor for medical purposes (MACvu, Marquette, Milwaukee, WI, USA; and Finometer, FMS system, Amsterdam, Netherlands). 

### 2.3. Signal Analysis

We followed recommendations to obtain accurate measurements of RR-intervals to analyze the smallest heart rate fluctuations [15]. Lead 2 of the electrocardiogram was digitized with a sampling frequency set at 500 Hz (AT-MIO-16, 12bits, Labview5.1, National Instruments, Austin, TX, USA). Intervals between R-peaks of the electrocardiogram were determined off-line by means of a peak detection algorithm. Electrocardiograms and time series of RR-intervals were visually inspected to identify R-peak misdetections and ectopic beats, which were manually deleted. Bradycardia sequences were identified on the time series of each patient, taking care not to include the large bradycardia of the syncopal episode and the preceding 30 s in cases of positive outcomes. A bradycardia sequence was defined as successive RR intervals with an increasing value. The length of a bradycardia sequence was defined as the total number of beats involved in the sequence. For each time series, bradycardia sequences were classified according to their length and were counted. The rank of bradycardia sequences of a same length was determined by classifying them according to their frequency of occurrence. For each patient, a diagram was plotted with the natural log of the rank on the x-axis and the natural log of the length of the matching bradycardia sequences on the y-axis. A linear regression was performed for each diagram in order to obtain the correlation coefficient and the slope. A previous study showed a cardiovascular Zipf’s distribution according to two straight lines with a tipping point [16]. In this study, the position of a tipping point was determined by the best linear fits for each diagram. 

### 2.4. Vasovagal Events

The method to assess and quantify the vasovagal events has previously been extensively described [4]. Vasovagal events were defined as consecutive beats with a drop in the mean blood pressure and an increase in the RR-interval. We classified and counted these events according to their length in number of beats.

### 2.5. Statistics

Data are presented as the mean ± SEM. Statistics were performed by means of Prism 5.01 (GraphPad Software, San Diego, CA, USA). We considered that the distribution fitted a straight line when |r| > 0.95. Tests for normality were performed by means of d’Agostino-Pearson omnibus K2 tests. Spearman correlations between Zipf’s and Gutenberg Richter’s law parameters were performed thanks to the data of a previously published study on the same data set focusing on this former law [4]. Matched patients with and without (pre)syncopal symptoms during the head-up tilt test (T+ and T−) were compared by means of a paired t test. We set the statistical significance at p < 0.05. 

## 3. Results

T+ and T− patients had comparable anthropomorphic characteristics, medical history, treatments, heart rate, and blood pressure, as previously reported (66 ± 1 vs. 67 ± 2 bpm, 131 ± 4 vs. 127 ± 4 mmHg, and 77 ± 4 vs. 74 ± 3 mmHg, heart rate, systolic, and diastolic blood pressure, respectively) [4]. Electrocardiography recordings were of a high quality, and the visual review identified only several false R peak detections. Ectopic beats were observed in only seven patients (three T+ and three T-) and were sparse (maximum of two per 5 min on two T- patients). The quality of the tachograms was good (Figure 1).

Bradycardia sequences were very frequent with no difference between T+ and T− and involved a large number of beats (36.2 ± 1.3 and 37.2 ± 1.1 beats per minute, respectively; Figure 2). Their maximal length was 12.1 ± 0.6 and 10.6 ± 0.6 beats for T+ and T−, respectively, with no difference between groups. 

Bradycardia sequences were distributed according to their rank along a straight line in all of the patients (T+ and T−). More precisely, they were distributed along two straight lines: one for the bradycardia sequences of a maximum of five beats and a second one for the longer bradycardia sequences with a coefficient correlation superior to 0.95 in all patients (T+ and T−; Figure 3). The position of the tipping point was the same in all patients. 

The slope of the relationship was significantly different between T+ and T− in the case of the long bradycardia sequences (Figure 4) but not in the case of shorter ones (−0.82 ± 0.05 and −0.78 ± 0.07 for T+ and T−, respectively; no unit; p = 0.686). 

The link between Gutenberg Richter’s and Zipf’s distributions was determined by comparing the slope of the linear relationship drawn by these two distributions. Gutenberg Richter’s distribution was assessed through the slope of the distribution of vasovagal sequences. Zipf’s distribution was assessed through the slope of the short and long bradycardia sequences. The slope values of the linear relationships were not normally distributed in cases of vasovagal and short bradycardia sequences (p < 0.0001 and p < 0.01, respectively), but were normally distributed in cases of long bradycardia sequences (p = 0.06). The slopes of the linear relationship of vasovagal and short bradycardia sequences were slightly but significantly correlated (r = 0.324, p = 0.02, Figure 5). There was no correlation between the slopes of the vasovagal and long bradycardia sequences’ relationship and between the slopes of short and long bradycardia sequences (r = 0.117, p = 0.441, and r = 0.111, p = 0.465, respectively, Figure 5, vasovagal sequence data are from [4]).

## 4. Discussion

This study confirms Zipf’s law of cardiovascular dynamics. It also shows that Zipf’s and Gutenberg Richter’s distributions provide complementary information despite a link between these two distributions. 

Several authors have dealt with Zipf’s law of cardiovascular dynamics by means of different approaches. To our knowledge, Kalda et al. were the first to demonstrate Zipf’s law of cardiovascular dynamics [10]. These authors studied the statistical similarities of short series of RR-intervals. Yang et al. performed an analysis based on logic variations of consecutive heart beats, while Rodriguez et al. looked for the presence of mathematical properties of Zipf’s series in RR-interval recordings [11,12]. The question remains whether these approaches used the most efficient way to assess Zipf’s law because the natural language of the cardiovascular system remains unknown. In our study, we began with a physiological observation to attempt to align as far as possible with this unknown language. This physiological observation is a conundrum. It is possible to observe bradycardia episodes on consecutive heart beats when the heart rate spontaneously fluctuates in a normal subject [17]. These episodes contradict the deterministic homeostatic regulation of the cardiovascular function. A fast regulatory mechanism called the baroreflex should detect blood pressure fluctuations and compensate beat-by-beat for these fluctuations by adjusting the heart rate. A decrease in the heart rate decreases blood pressure, but the baroreflex should increase the heart rate to stop the blood pressure decrease. A bradycardia episode of several beats is not expected in cases of a well-working deterministic homeostatic baroreflex. A prolonged bradycardia episode could eventually lead to large arterial hypotension with compromised brain perfusion and loss of consciousness. The phenomenon is called vasovagal syncope and could paradoxically occur in any normal subject despite a well-working baroreflex [6]. In this study, we tried to stay close to the natural physiological language of the cardiovascular system. This approach allowed us to define a simple method with strong evidence of Zipf’s law in the cardiovascular dynamics. However, further studies may help to better define the natural cardiovascular language to better characterize Zipf’s law and the self-organized properties of the cardiovascular function.

Our approach also differs from the previous studies on Zipf’s law in the cardiovascular system. Heart rate variability recordings were obtained by means of a Holter monitor in all of the three previous studies, while we studied quiet unmoving patients [10,11,12]. A Holter monitor is a device that records the heart rate during the normal daily life of the patient. The heart is constantly influenced by the various demands of daily life that include activities, stressors, and body position. On Holter recordings, heart rate variability is the result of daily life but is also intrinsic to regulatory mechanisms and their physiological delays. Daily life variability is totally absent in immobile and quiet patients, and solely the intrinsic variability remains. We previously demonstrated the influence of differences in experimental set-up in analyses of heart rate variability, including those with a focus on its complex dynamics [18,19].

The definitions of the sequences to study Gutenberg Richter’s and Zipf’s laws both included bradycardia episodes on consecutive heart beats. This point therefore identifies overlap between the meaning of Gutenberg Richter’s and Zipf’s distributions with these two close definitions. Thus, the slopes of Gutenberg Richter’s and Zipf’s distributions are correlated. However, only the short bradycardia sequences are correlated with vasovagal sequences and not the long ones. Moreover, the distribution of long bradycardia sequences differs between T− and T+ patients, contrary to Gutenberg Richter’s distribution (Figure 4). This difference shows that Gutenberg Richter’s and Zipf’s laws provide complementary information about cardiovascular self-organized criticality.

Telling the difference between patients with and without a positive result in the diagnosis tool for vasovagal syncope is a challenge of cardiovascular medicine. Patients with vasovagal syncope are usually apparently healthy after a regular medical check-up [6]. The baroreflex is functioning well in these patients, who generally maintain their cardiovascular function well. Vasovagal syncope has remained a medical mystery for centuries [6]. Only recently, some studies focusing mainly on the complex dynamics of heart rate variability convincingly showed a difference between patients with a positive outcome of the diagnosis tool and patients with a negative one. Graff et al. demonstrated this difference by means of the entropy method, while Fortrat et al. achieved this by defining a marker of cardiovascular instability [20,21]. Questions remain about whether these two studies and Zipf’s law finally focused on the same complex properties by means of different tools or whether the different methods provide complementary and unrelated information.

The main limitation of this study is the analysis of the supine and standing parts of the head-up tilt table test into single time series. The standing position requires major cardiovascular adaptions partly driven by the autonomic nervous system [22]. Cardiovascular dynamics and heart rate variability are different between these two positions, as demonstrated by means of signal analysis or only by looking at the time series (Figure 1). We previously demonstrated the influence of body position on the cardiovascular Zipf’s distribution [16]. The analysis of a single time series for a whole head-up tilt table test was however necessary to collect enough of the sparse vasovagal events in order to perform Gutenberg Richter analysis [4]. Our study demonstrated that the focus can be placed on the heart rate to study cardiovascular self-organized criticality. Specifically, future studies should clarify the influence of body position adaptation on cardiovascular self-organized criticality and investigate the influence of autonomic nervous system adaptations on SOC. Future studies should also confirm that the tipping point of the bradycardia Zipf’s distribution is linked to physiological influences like breathing and not to analysis artifacts like a finite size effect. 

## 5. Conclusions

This study confirmed Zipf’s law of cardiovascular dynamics and demonstrated that Zipf’s and Gutenberg Richter’s laws explore complementary aspects of the self-organized criticality of cardiovascular dynamics. Zipf’s law provides an interesting and easy to implement tool to better characterize the self-organized criticality of cardiovascular dynamics. Zipf’s law may also provide an interesting tool for the medical diagnosis of some cardiovascular diseases.

## Figures and Tables

**Figure 1 entropy-22-00413-f001:**
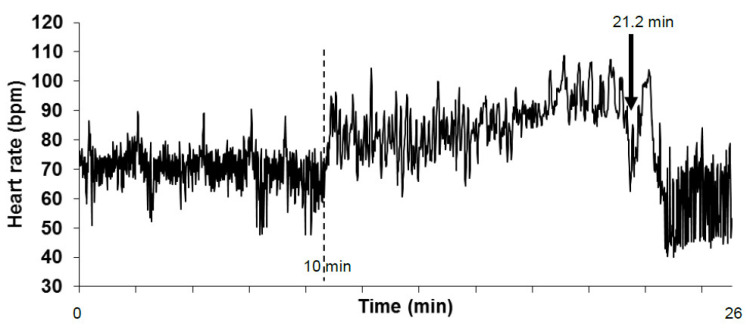
Tachogram of a patient obtained during a head-up tilt table test. The tachogram is the beat-by-beat heart rate plotted against time (y axis and x axis, respectively). The first part of the tachogram is obtained in the supine position and ends at the vertical dashed line (10 min). The second part of the tachogram is the head-up position. The head-up position was stopped at the (pre)syncope occurrence (arrow). The analyzed time series started at the beginning of the tachogram and ended before the (pre)syncope occurrence, so it was excluded. The heart rate is shown here in beats per minute for convenience but is measured as RR-intervals (in ms) on the electrocardiographic signal. (bpm: beats per minute).

**Figure 2 entropy-22-00413-f002:**
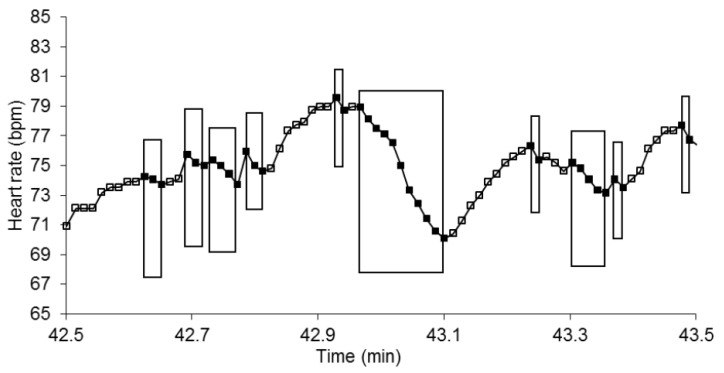
One minute of a patient’s heart rate over time (y and x axes, respectively). Each heart beat is indicated by a square. Each box indicates a bradycardia sequence. The heart rate is shown in beats per minute for convenience but is measured as RR-intervals (in ms) on the electrocardiographic signal. (bpm: beats per minute).

**Figure 3 entropy-22-00413-f003:**
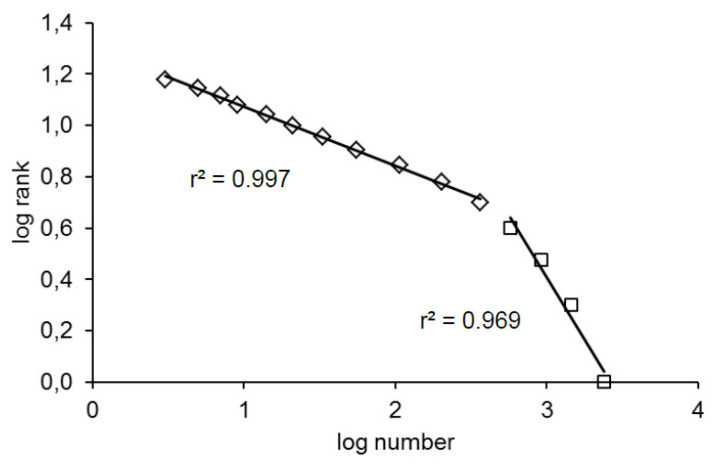
Distribution of the number of bradycardia sequences according to their rank in one patient (log-log plot in natural logarithm). The pattern is the same for all patients including the position of the tipping point.

**Figure 4 entropy-22-00413-f004:**
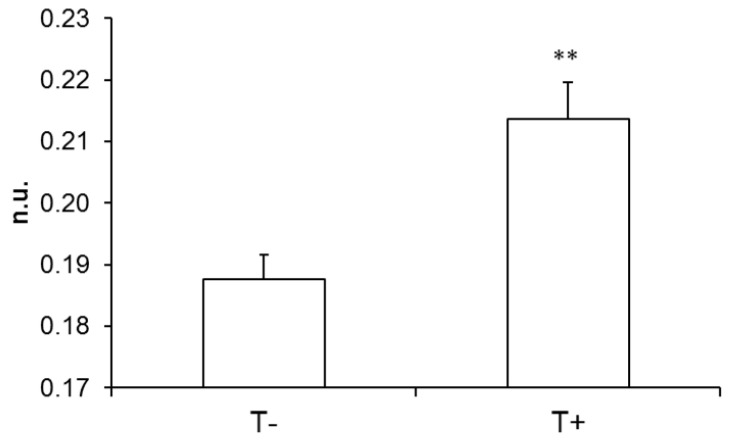
Absolute value of the slope of Zipf’s distribution of bradycardia sequences longer than five beats in a group of patients with a negative outcome to the head-up tilt test and a matched group of patients with a positive outcome (T− and T+ respectively; **: p < 0.01). n.u.: no unit.

**Figure 5 entropy-22-00413-f005:**
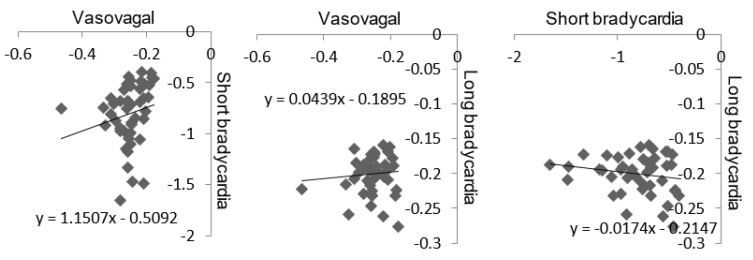
Link between Gutenberg Richter’s and Zipf’s distribution according to the slope of the linear relationship drawn by these two distributions. Gutenberg Richter’s distribution was assessed through the slope of the distribution of vasovagal sequences (axis label: Vasovagal). Zipf’s distribution was assessed through the slope of the short and long bradycardia sequences (axis label: Short bradycardia and Long bradycardia, respectively). The coefficients of correlation and significance are mentioned in the text.

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
