# Peer review of "Zipf’s Law of Vasovagal Heart Rate Variability Sequences"

_entropy, 2020, doi:10.3390/e22040413_

Round 1

Reviewer 1 Report

The resubmitted manuscript addressed properly the critical issues raised in the first revision. However, the author should consider the following minor points:

Line 114. There is a typing error in the sub-heading title (2.4 Vasovagal events).

Line 117. "We these events": the verb is missing.

Figure 1. It is useless to indicate that the time is expressed in minutes if the horizontal axis does not show any label, in any units. Please add at least the labels corresponding to the minimum (0' ?) and maximum (30' ?) values of the axis.

Figure 3. Please add the information on the logarithm base also in the figure legend, e.g., "log-log plot in natural logarithm".

Line 251. The citation (Fortrat and Ravé) should be indicated as [22].

Author Response

I thank the reviewer for the positive comments.

>Line 114. There is a typing error in the sub-heading title (2.4 Vasovagal events).

I fixed the problem.

>Line 117. "We these events": the verb is missing.

The phrase now includes verbs.

>Figure 1. It is useless to indicate that the time is expressed in minutes if the horizontal axis does not show any label, in any units. Please add at least the labels corresponding to the minimum (0' ?) and maximum (30' ?) values of the axis.

The label of the minimum (0) and maximum (26) are now mentioned.

>Figure 3. Please add the information on the logarithm base also in the figure legend, e.g., "log-log plot in natural logarithm".

I added this mention.

>Line 251. The citation (Fortrat and Ravé) should be indicated as [22].

The citation is now correctly mentioned [16].

Reviewer 2 Report

I have now studied the revised version of the manuscript "Zipf's law of vasovagal heart rate variability sequences".   I thank the author for addressing my comments. While I keep my opinion that some terms used in this manuscript are likely to be misunderstood by the researchers working complexity systems, the results are interesting and deserve publication in Entropy. I thus recommend publication in present form.

Author Response

I thank the reviewer for the positive comments.

This manuscript is a resubmission of an earlier submission. The following is a list of the peer review reports and author responses from that submission.

Round 1

Reviewer 1 Report

The manuscript "Zipf's law of vasovagal heart rate variability sequences" by Fortrat presents an investigation about the size of bradycardia sequences. This quantity is defined as the length of downtrend sequences in time series of heart rate (Figure 2). The author describes the results of 48 patients grouped according to the outcome of head-up tilt test (24 with positive outcomes and syncope symptoms). The results depict a rank plot for one patient, where the author fits the curve with two exponents. The author also argues that the exponents for long sequences are different between patients with positive (T+) and negative (T-) outcome of head-up tilt tests.

I believe the results are interesting, but there are some issues the author needs to address before publication.

1) The author needs to be more careful with the use of the term "complexity" over the manuscript. The use of this term in the opening phrase "Complexity, the final frontier of the cardiovascular system, has emerged as a major topic over the last decade." and in many others is very vague. I suggest the author rephrase these sentences by "complexity science approaches" and "cardiovascular complexity". Similarly, the use of self-organized criticality (SOC) is also quite vague. An *indicative* of power-law behavior is not enough for characterizing SOC as defined by Per Bak. I thus suggest the author avoid this term over the paper.

2) I also believe the author is somehow misinterpreting the Gutenberg-Richter (GR) law and its connection with the Zipf law. The GR law states that number of earthquakes N with magnitude larger than M is modeled by log N ~ -b M. If we assume that the magnitude of earthquakes is distributed according to P(M), then, N ~ \int_M^\infty P(M')dM', which implies P(M) ~ 10^(-b M). This probability distribution decays *exponentially* with M. In case of earthquakes, energy E and magnitude are related via log E = A + B M. By using this relationship and P(M), we can find the probability distribution for the energy P(E) that is P(E) ~ E^(-\beta), with \beta=b/B -1. Thus, the *energy* of earthquakes is power-law distributed. Because there is no such thing as a "sequence magnitude", the use of the term GR law is quite odd and should be avoided. Furthermore, if the author is referring to the GR law as a power-law (cumulative) distribution for the sequence length, the Zipf exponent and power-law exponent are the same. This somehow can be verified by comparing the results of Figure 3 with those in Figure 3 of Ref. 5. If that is the case, then statements such as "This study confirms Zipf's law of cardiovascular dynamics. It also shows that Zipf’s and Gutenberg Richter's distributions provide complementary information despite a link between these two distributions." are meaningless. I suggest the author have a look into this text https://www.hpl.hp.com/research/idl/papers/ranking/ranking.html

3) The author needs to describe better the approach used to fit the two lines over the Zipf plots. For instance, how the tipping point is estimated? Does it change among different patients? In general, I am not convinced about the significance of the second power-law regime. I think this is more likely to be related to finite size effects and may be better described by an exponential cutoff. It is also essential to show the results for more patients.

Author Response

AU: I appreciated the reviewer’s comments that demonstrate a careful and deep reading of our manuscript. Thank to these comments I greatly improved our manuscript. I think that the reviewer will be satisfied by the new version of our manuscript. You will find below an answer to all of the points raised by the reviewer.

RW1: 1) The author needs to be more careful with the use of the term "complexity" over the manuscript.

AU: I agree with the reviewer: The term “complexity” is quite vague. Accordingly, the term “complexity” is no more associated with “heart rate variability”. I use the term “complex dynamics of heart rate variability” in the new version of the manuscript. However, I have been surprised by this comment. The term “complexity” is commonly used in the context of cardiovascular dynamics and more generally in physiology (see the review Shaffer F, Ginsberg JP. Front Public Health. 2017 Sep 28;5:258). Does it possible that this term is not commonly used in the field of the reviewer? Complexity is also largely used in multidisciplinary approaches of the sciences of complexity. This term appears in book titles and I will give only one: “Worlds Hidden in Plain Sight: The Evolving Idea of Complexity at the Santa Fe Institute, 1984–2019” (D Krakauer editor). Accordingly, I have maintained the term “complexity” in the beginning of the introduction until L35 when it is used in a broad sense.

RW1: Similarly, the use of self-organized criticality (SOC) is also quite vague. An *indicative* of power-law behavior is not enough for characterizing SOC as defined by Per Bak. I thus suggest the author avoid this term over the paper.

AU: The reviewer probably wants me to avoid the term “self-organized criticality” because of the misunderstanding raised on comment #2. I hope that answer to this comment #2 will satisfy the reviewer. SOC is central to this work that is fully inspired by Bak’s works. Deleting the term SOC over the manuscript is equivalent to gut it. I would prefer withdraw the manuscript rather than deleting this term.

RW1: 2) I also believe the author is somehow misinterpreting the Gutenberg-Richter (GR) law and its connection with the Zipf law. The GR law states that number of earthquakes N with magnitude larger than M is modeled by log N ~ -b M. If we assume that the magnitude of earthquakes is distributed according to P(M), then, N ~ \int_M^\infty P(M')dM', which implies P(M) ~ 10^(-b M). This probability distribution decays *exponentially* with M. In case of earthquakes, energy E and magnitude are related via log E = A + B M. By using this relationship and P(M), we can find the probability distribution for the energy P(E) that is P(E) ~ E^(-\beta), with \beta=b/B -1. Thus, the *energy* of earthquakes is power-law distributed. Because there is no such thing as a "sequence magnitude", the use of the term GR law is quite odd and should be avoided.

AU: I think that this comment is the result of a misunderstanding about the clinical presentation of vasovagal syncope. The manuscript was probably unclear and I rewrote the description of vasovagal syncope (L41-43 and see below).

There is a thing such as magnitude in case of vasovagal syncope: a vasovagal episode could be of very low magnitude with no symptoms at all, it could also lead to self-limiting symptoms, or to symptoms that impose to interrupt the standing position, or symptoms with fall and loss of consciousness, the postictal recovery could be quick or slow with a deep asthenia, the loss of consciousness might be due to bradycardia of different intensities including heart stop of several seconds, of several tens of seconds. You can quantify the “magnitude” of a vasovagal syncope by quantifying the intensity of symptoms, the deepness of the bradycardia, the deepness of the hypotension, the delay of recovery or like in our peer reviewed paper ref #4 the length of the vasovagal reaction… most of the patients spontaneously mention the felt “magnitude” of the spells they report. In order to introduce the “vasovagal magnitude” to the reader not familiar with vasovagal syncope, I rewrote the description of vasovagal syncope in the introduction. Accordingly, vasovagal syncope perfectly match earthquakes.

The reviewer is however right: “there is no such thing as a “sequence magnitude”” but only in the case of bradycardia sequences. All the vasovagal sequences include a bradycardia but some bradycardias are not involved in a vasovagal sequence. However, bradycardias are remarkable sequences of beats like words are remarkable sequences of letters.

Accordingly, vasovagal sequences match earthquakes and bradycardia sequences match words.  

RW1: Furthermore, if the author is referring to the GR law as a power-law (cumulative) distribution for the sequence length, the Zipf exponent and power-law exponent are the same. This somehow can be verified by comparing the results of Figure 3 with those in Figure 3 of Ref. 5.

AU: I think that the reviewer would like to mention the Figure 3 of Ref 4 because I do see any link between Figure 3 of the submitted manuscript and Figure 3 of Ref 5. A goal of our study as stated in the introduction is to perform a comparison between GR and Zipf distribution (while GR and Zipf distribution are not drawn from the same data). Indeed this study actually compared the results of Figure 3 with those in Figure 3 of Ref 4. Results are mentioned in the Results section L180 to L190. However, the reviewer #2 mentioned that the phrasing of these lines has to be improved in order to make clearer the link to this comparison. This poor phrasing probably explains the reviewer’s comment. I rewrote these lines L180-L190.

RW1: 3) The author needs to describe better the approach used to fit the two lines over the Zipf plots. For instance, how the tipping point is estimated? Does it change among different patients? In general, I am not convinced about the significance of the second power-law regime. I think this is more likely to be related to finite size effects and may be better described by an exponential cutoff.

AU: The reviewer identified a weakness of the manuscript. The method to estimate the tipping has to be mentioned. I have been largely influenced by other ongoing experiments about cardiovascular SOC but the reader could not be aware about these not yet published data. One of these experiments is now accepted as an abstract at the ESGCO meeting in Pisa. I added this reference in the new version of the manuscript [16]. I now mention in the new version of the manuscript that all the patients have the same Zipf distribution (legend of Figure 3, and L163-164). I now mention in the new version of the manuscript that the position of the tipping point was determined by the best linear fits (L120-121). In the first version of the manuscript, I discussed only the physiological potential explanations for the tipping point and definitively forgot to mention possible analysis artifact as suggested by the reviewer. I now mention this point in the new version of the manuscript (L265-267)

RW1: It is also essential to show the results for more patients

AU: I now mention in the new version of the manuscript as well as in the figure legend that all the patients have the same Zipf distribution. (Legend of Figure 3, and L163-164).

Reviewer 2 Report

The authors present a statistical analysis of cardiovascular dynamics.
Thay analyse heart rate variability recordings obtained by means of a Holter monitor for quiet unmoving patients.
In this way they eliminated the daily live heart-rate variability and extracted the intrinsic heart-rate variability.
They showed that this dynamics fulfill Gutenberg Richter’s and Zipf’s law. This provide complementary information about cardiovascular self-organized criticality.
This seems to be an interesting way to study of cardiovascular self-organized criticality and may also provide an interesting tool for medical diagnosis of diseases related to heart rhythm.
But it certainly requires more cases to be analyzed.

I recommend this article for publication in a present form.

Author Response

No more comments

Author Response

AU: I appreciated the reviewer’s comments that demonstrate a careful and deep reading of our manuscript. Thank to these comments I greatly improved our manuscript. I think that the reviewer will be satisfied by the new version of our manuscript. You will find below an answer to all of the points raised by the reviewer.

RW3: 1) Introduction. At lines 52-56 the author states that the second goal is to evaluate whether the Zipf law in heart rate and the Gutenberg Richter’s law in heart rate and blood pressure provide similar information. Most of the readers are not familiar with these laws and here the author should briefly mention the definition of these two laws and why differences between them could be expected.

AU: I now mention in the new version of the manuscript a definition of these two laws (L38-39, and L56-57).

RW3: 2) Patients. It seems that the two groups of participants were selected from a larger database to be compared as matched groups. Even if the same participants were described previously, the author should provide some details on the larger original database, on the exclusion criteria and on the way T+ and T- were matched.

AU: I now mention the anthropomorphic characteristics of the primarily included patients (L74). I describe the flow chart mentioning the exclusion criteria (L72-88). I now mention more clearly that the T+ and T- were matched according to the age and sex (L82).

RW3: 3) Signal analysis. The ECG sampling rate was 500 Hz. If the R peak is not interpolated to refine the time of occurrence of the maximum, the resolution is 2 ms, larger errors affect the calculation of the R-R intervals and even larger errors may affect the difference between consecutive R-R intervals (DeltaRRI). Thus, it is possible that a small positive DeltaRRI is actually evaluated as a null increment (DeltaRRI=0 ms), underestimating the true length of the bradycardic sequence. Are the bradycardic sequences defined by positive increments only, or they may also include null increments? What is the minimum positive DeltaRRI that can be detected?

AU: The ECG sampling rate is critical for diagnosis. There are expert consensuses about this sampling rate. I now mention that the sampling rate of 500 Hz is in accordance with the usual recommendations (L104-105). I mention the Rompelman’s paper about this point [15].

RW3: 4) Results. Line 112-113. For clarity, some general characteristics, as well as the mean heart rate levels, should be indicated here even if already reported in ref.[4].

AU: Anthropomorphic characteristics of T+ and T- are now mentioned in the Material and Method section (L85-86). Heart rate and blood pressure are mentioned in the Results section (L138-139).

RW3: 5) Results. Line 126-129. One of the declared limitations of the study is that it considered data segments with different characteristics: supine (SUP) and head-up tilt (HUT). Therefore, a general description of the SUP and HUT segments separately should be provided. For instance, is the maximal length of the bradycardic sequence equal to 12 beats also in SUP for the T+ groups? Is the number of beats in the bradycardic sequence equal to 36 in HUT and in SUP similarly? The author may consider including these data in a table that summarizes the comparison between T+ and T- not only for the whole recording but also for SUP and HUT separately, adding the corresponding p values. In fact, information on the maximal length of the bradycardic sequences in SUP could be useful to reinforce the sentence of the discussion section (lines 222-3) on the influence of body position and to add a comment on the possibility to distinguish the instability leading to the vasovagal syncope also from supine recordings.

AU: The reviewer perfectly understood that the key to understand and explain the pathophysiology of vasovagal reaction is the physiology of standing SOC cardiovascular dynamics. Studying this physiology on vasovagal patients is however not a good idea and is full of pitfalls. Standing cardiovascular dynamics needs time to adapt from the position shift. This time is about 30 s for heart rate but could be much longer for other variables. This standing cardiovascular dynamics is also probably altered before syncope occurrence. Time of syncope is different from patients to patients. However, the reviewer is right: the physiology of standing SOC cardiovascular dynamics is a fascinating topic. I mention in the new version of the manuscript a now accepted abstract at ESCGO Pisa 2020 about this topic [16].  

RW3: 6) Figure 3. The example of figure 3 should be better described because, as it stands now, it is rather confusing. First, what is the base of the log function? If the base is 10, as in the author’s previous work [4], then the longest rank corresponds to more than 10^3=1’000 beats! Did the author use the natural logarithm instead? For clarity, the figure legend should report the numerical values for the shortest and for the longest ranks represented in this example and for the corresponding number of sequences.

AU: The reviewer identified a mistake in this Figure 3. The label of the x and y axis was inverted. Thanks to the reviewer, I fixed this problem that explains comment #6, #7, and #8.

RW3: 7) Figure 3. Second, since log(1)=0 and the logarithm of the lowest rank in fig.3 is >0, it seems that the length of the shortest possible sequence is >1. If this is the case, then the definition of the length of a bradycardia sequence should be clarified in lines 95-97. In fact, when the author wrote that “A bradycardia sequence was defined as successive RR intervals with an increasing value” and that the sequence length is the number of their beats, one understands that the shortest sequences (as the last one depicted in figure 2) have length of one beat, being defined by only one RR interval with an increasing value. Please clarify.

8) Figure 3. Third, the author wrote that the Zipf’s law was represented by two straight lines on the log-log plots, one of which “for the bradycardia sequences of a maximum of five beats”. By contrast, the example of figure 3 suggests that the shorter straight line includes sequences with length from 2 beats up to 12 beats. Was the shortest slope evaluated only for lengths up to 5 beats, in contrast with the example of figure 3? What is the range of the ranks for evaluating the long-term slope?

9) Results. line 145. Please indicate the p-value also for the nonsignificant difference between T+ and T-.

RW3: 10) Results. line 155-157. It is very difficult to understand that the slope of vasovagal sequences mentioned here and correlated with the slopes of the bradycardia sequences is the slope of the power-law represented by the Gutenberg Richter’s law. In fact, a definition of the Gutenberg Richter’s law is missing. The author should recall in methods how the Gutenberg Richter’s slopes were calculated for the vasovagal episodes in ref.[4].

AU: I now added the definition of the vasovagal events. They are defined in the §2.4 Vasovagal sequences in the Material and Methods section (L122-126). I rephrased the result section in order to clarify that this section is about the comparison between GR and Zipf distribution (L180-190).

RW3: 11) Results. Moreover, since the evaluation of the common information provided by the Gutenberg-Richter’s law and by the Zipf’s law is one of the main goals of the study, the author should consider not to just report the “r” and “p” of the linear correlation, but to also plot the whole cloud of points representing the associations between short- (or long-) bradycardic slopes and the vasovagal slopes. The absence of a linear correlation does not mean that an association (possibly nonlinear) does not exist.

AU: I added Figure 5.

RW3: Minor

Figure 1. Please add the duration of the horizontal time axis.

AU: The duration was mentioned but probably not clearly enough. I added time labels for two main steps of the test: position shift from supine to head-up and syncope occurrence.

RW3: line 53 “They show a linear distribution…”. Who are “they”? Does the author mean “These previous works show…”?

AU: This was unclear. I rewrote (L53).

RW3: Discussion line 163. “Several authors have dealt with Zipf’s law”. This statement seems in contrast with the introduction reporting that “This law has been incidentally described…”

AU: I delete the word incidentally in the new version of the manuscript (L57).

RW3: Abstract. I am not sure that the last conclusion “Bradycardia distribution could provide an interesting diagnosis tool for heart rhythm diseases” is properly supported by the results on the vaso-vagal syndrome. A more accurate conclusion could be that “Bradycardia distribution could be a tool for detecting cardiovascular instability as in the vaso-vagal syndrome”.

AU: The term “instability” is quite confusing in this case and should be avoid if not previously defined in the manuscript. Vaso-vagal syncope is not a syndrome. I suggest “Zipf’s law may also provide an interesting tool for medical diagnosis of some cardiovascular diseases” (L22 and L274).